# Coronavirus Disease 2019-Associated Coagulopathy

**DOI:** 10.3390/microorganisms10081556

**Published:** 2022-08-02

**Authors:** Jun-Won Seo, Da-Young Kim, Nara Yun, Dong-Min Kim

**Affiliations:** Division of Infectious Disease, Department of Internal Medicine, College of Medicine, Chosun University, Gwangju 61453, Korea; jwseo83@chosun.ac.kr (J.-W.S.); dayz02@hanmail.net (D.-Y.K.); shine-0222@hanmail.net (N.Y.)

**Keywords:** COVID-19, coagulopathy, hypercoagulability, thrombocytopenia

## Abstract

Coronavirus disease 2019 (COVID-19)-associated coagulopathy is an acute illness characterized by thrombosis with or without hemorrhage after COVID-19 infection. Clinical symptoms of COVID-19-associated coagulopathy can occur at any anatomical site. Various forms of venous thromboembolism, including deep vein thrombosis and pulmonary embolism, are common in acutely ill patients with COVID-19. Laboratory findings, such as D-dimer and platelet counts, can help diagnose COVID-19-associated coagulopathy. Anticoagulation using direct oral anticoagulants and low-molecular-weight heparin is essential for the treatment of COVID-19-associated coagulopathy. Prophylactic anticoagulants are important in preventing COVID-19-associated coagulopathy in patients with severe COVID-19. In particular, the early initiation of prophylactic anticoagulation in patients with COVID-19 can improve survival rates without the risk of serious bleeding events.

## 1. Introduction

In December 2019, a novel coronavirus was identified as the cause of the pneumonia clusters emerging from Wuhan, China. The virus causative pathogen responsible for coronavirus disease 2019 (COVID-19) was identified as severe acute respiratory syndrome coronavirus-2 (SARS-CoV-2), which quickly spread worldwide, making us live in the COVID-19 pandemic era. Patients with COVID-19 present with various clinical manifestations and complications. Although respiratory symptoms are a major characteristic problem of COVID-19, some patients had complex hypercoagulable abnormalities that are closely related to mortality [1,2]. Numerous pathogenic mechanisms may be involved in COVID-19, and evidence suggests that COVID-19 is associated with clotting dysfunction that increases the risk of venous and arterial thromboembolism (TE), which increases the risk of death. The incidence of TE in patients with COVID-19 has been reported in various studies [3,4,5]. Some studies have reported a low prevalence of TE in COVID-19. The prevalence is less than 10%, especially for patients who are outpatients or admitted to non-intensive care units (ICU) who have mild disease. On the other hand, the incidence of TE in patients with severe COVID-19 exceeds 20%, and patients with severe COVID-19 who are admitted to the ICU, in particular, show a very high incidence of more than 30%. Some studies have reported a high rate of close to 70% [5,6,7]. In this review, we describe the characteristics of COVID-19-associated coagulopathy (CAC) and discuss the latest strategies to reduce death due to thrombosis in patients with confirmed COVID-19.

## 2. Pathogenesis

However, the pathogenesis of hypercoagulability in COVID-19 remains unclear. Generally, thrombosis can develop when there are three main preconditions, called Virchow’s triad: endothelial injury, stasis of blood flow, and a hypercoagulable state [8]. Similarly, these three major contributions to clot formation can be applied to severe COVID-19 infections. Figure 1 shows the pathophysiology of CAC. Coagulation begins with the binding of SARS-CoV-2 to the angiotensin-converting enzyme 2 (ACE2) receptor on endothelial cells, which induces systemic and vascular inflammation (including increased fibrinogen and interleukin [IL]-6), exposes tissue factor (TF) and collagen to the blood, and releases von Willebrand factor (vWF) from Weibel–Palade granules. Platelets are then exposed and activated by TF, collagen, and vWF to release adenosine monophosphate (ADP). In addition, vWF further induces platelet recruitment, leading to activation, aggregation, and plug formation. The TF-factor VII interaction activates the extrinsic pathway, and the exposed collagen initiates the intrinsic pathway. Both pathways create a common coagulation pathway, which ends with the formation of fibrin strands, resulting in the formation of stable platelet-fibrin clots. In addition, the ACE2 receptor is down-regulated by the binding of SARS-CoV-2, which increases angiotensin 2 levels, resulting in angiotensin 2-induced coagulopathy [9]. Angiotensin 2 activates platelets expressing P-selectin and cluster of differentiation 40 ligand (CD40L) and binds to CD40 and P-selectin glycoprotein ligand-1 (PSGL-1) expressed in neutrophils to form neutrophil extracellular traps (NETs), which have strong prothrombotic properties and inactivate thrombomodulin. Angiotensin 2 contributes to thrombogenesis by inducing the expression of plasminogen activator inhibitor-1 (PAI-1), intercellular adhesion molecule-1 (ICAM-1), and vascular cell adhesion molecule-1 (VCAM-1) in endothelial cells through the angiotensin II receptor (AT1R), and by inducing the expression of TFs in monocytes with P-selectin. In addition, immobility by mechanical ventilation and high positive end-expiratory pressure, which reduce blood flow in venous return, induce venous stasis. As a result, the Virchow triad is completed, and COVID-19-related thrombosis occurs through this series of processes.

Several studies reported that SARS-CoV-2 is contributed to the direct invasion of endothelial cells, leading to endothelial injury [10,11]. Endothelial cell dysfunction is an important mechanism of the pathological pathway in CAC to the same degree as the cytokine storm. The main pathophysiological results of endothelial cell dysfunction are hypercoagulation and angiogenesis. Norooznezhad et al. reported that endothelial cell dysfunction-induced hypercoagulation could be caused by alterations in the levels of PAI-1, vWF antigen, soluble thrombomodulin, and tissue factor pathway inhibitor (TFPI) [12]. Complement-mediated endothelial injury is also possible through activation of the alternative complement pathway by the spike protein of SARS-CoV-2 [13,14]. Cugno et al. also found that patients with COVID-19 had high plasma markers of complement activation and endothelial dysfunction, such as soluble C5b-9, C5a, vWF, tissue plasminogen activator (tPA), and PAI-1 [15]. Furthermore, they showed that soluble C5b-9 and vWF levels were correlated with disease severity. Stasis of blood flow can be sufficiently induced in critically ill patients with COVID-19 who have limited movement due to mechanical ventilation or extracorporeal membrane oxygenation (ECMO). A hypercoagulable state can occur in patients with COVID-19 due to changes in prothrombotic factors, such as elevated factor VIII, fibrinogen, and D-dimer [16,17]. Maier et al. reported 15 patients with severe COVID-19 who showed hyperviscosity as a result of capillary viscometry and estimated that hyperviscosity promotes hypercoagulation [18]. Robba et al. found that the mechanism of activation of the coagulation cascade in COVID-19 is the TF pathway, which causes endotoxin and tumor necrosis factor (TNF)-mediated production of IL and platelet activation. The consequent massive infiltration of activated platelets may be responsible for inflammatory infiltrates in the endothelial space, as well as thrombocytopenia [19]. Eslamifar et al. showed that overproduction of proinflammatory cytokines, such as IL-6, IL-1β, and TNF-α, induces cytokine storms and increases the risk of clot formation, platelet activation, and multiorgan failure [20]. Furthermore, they suggested that monocytes and macrophages infected with SARS-CoV-2 directly release TFs that activate the coagulation cascade, which may play an important role in the development of COVID-19 coagulation. Lazzaroni et al. reported that SARS-CoV-2 causes coagulation dysfunction, a virus-specific mechanism related to the virus interaction with the renin–angiotensin system (RAS) and the fibrinolytic pathway [21]. The authors found that the virus-mediated engagement of ACE2 decreases its expression and activates the RAS, promoting platelet adhesion and aggregation. High levels of PAI-1, acting as a major inhibitor of fibrinolysis interfering with tPA and urokinase, have been associated with an increased risk of thromboembolic events. The predominant coagulation abnormalities in patients with COVID-19 suggest a hypercoagulable state distinct from disseminated intravascular coagulopathy (DIC) and vaccine-induced immune thrombotic thrombocytopenia (VITT) (Table 1). Generally, DIC has an associated bleeding tendency that follows from the secondary activation of fibrinolysis, whereas the major clinical finding in COVID-19 is thrombosis [22]. In addition, DIC is characterized by the presence of markedly elevated fibrin degradation products (FDP), as seen in COVID-19, including a marked increase in D-dimer levels. However, other coagulation parameters such as fibrinogen, vWF, and factor VIII activity in COVID-19 are distinct from those in DIC [23,24]. Thus, the hypercoagulable state in patients with COVID-19 is more similar to compensated DIC than that of acute DIC. Marked elevation of D-dimer and FDP can be seen in both DIC and COVID-19, and these results are strongly associated with the mortality of COVID-19 [1,25]. In contrast, prolongation of prothrombin time (PT) and activated partial thromboplastin time (aPTT) in patients with COVID-19 is only moderate, and the findings of an increase in both fibrinogen and factor VIII can be interpreted as a more typical finding of response to acute infection of COVID-19 than that of DIC [17]. Although thrombocytopenia is also considered a risk factor associated with death, the degree of thrombocytopenia observed in patients with COVID-19 is lower than that commonly observed in DIC [26,27].

## 3. Clinical Manifestations

CAC can occur in multiple organs of the body, presenting in the form of thrombosis in conditions such as cerebrovascular accident, myocardial infarction, limb or mesenteric ischemia, deep vein thrombosis (DVT), and pulmonary embolism (PE) [28]. Xu et al. analyzed the SARS-CoV-2 nucleic acid using real-time RT-PCR of the serum and throat swab samples for evaluation of the relationship between SARS-CoV-2 RNAemia and organ damage such as respiratory failure, heart damage, kidney damage, and coagulation abnormality in patients with COVID-19 [29]. They reported that SARS-CoV-2 can damage multiple organs in association with SARS-CoV-2 RNAemia. In addition, they confirmed that the patients with RNAemia had high D-dimer and reported that the induction of these procoagulation factors can cause thrombosis in patients. The thromboembolic manifestations of CAC are extensive and appear to vary widely among clinical studies. Venous thromboembolism (VTE), including DVT and PE, is common in acutely ill patients with COVID-19. It has been confirmed that up to one-third of the patients with severe COVID-19 were admitted to the ICU. In some cases, TE has been confirmed, even though preventive anticoagulant therapy is being applied [30,31,32]. Autopsy studies have shown that hypercoagulability in patients with COVID-19 contributes to mortality. Mentor et al. reported that in post-mortem examination of patients with COVID-19, prominent PE was identified in 4 out of 21 (19%) patients, and alveolar-capillary microthrombosis was found in 5 out of 11 patients (45%) [33]. Of these, 11 deaths were confirmed to be due to TE despite receiving any form of anticoagulant treatment prior to death. Wichmann et al. reported that DVT was found in 7 of 12 patients (58%) who died of COVID-19 [34]. All seven patients had DVT in both legs, and no patient had a history of DVT or suspected DVT prior to death. In addition, 5 of 12 patients (42%) had PE findings, of which 4 had a direct cause of death. Anticoagulation treatment before death was administered to 4 out of 12 patients. Ackermann et al. compared the lung pathology of seven patients who died from COVID-19 with that of a control group who died from influenza or other diseases [35]. They reported a marked increase in endothelial damage, thrombosis with microangiopathy, and alveolar-capillary microthrombosis in the lungs of patients who died from COVID-19 compared with the lungs of patients who died from other causes. Bilaloglu et al. analyzed more than 3000 hospitalized patients with COVID-19 [36], most of whom received prophylactic anticoagulation, and the risk factors for VTE were old age, male sex, race (Hispanic), coronary artery disease, prior history of myocardial infarction, and high D-dimer levels (>500 ng/mL) upon admission using multivariate analysis. In addition, thrombotic events were independently associated with mortality (adjusted hazard ratio [HR], 1.82; 95% confidence interval (CI), 1.54–2.15; *p* < 0.001). They also reported arterial thrombosis, including stroke (1.6%) and myocardial infarction (8.9%). Old age, male sex, race (Hispanic), history of coronary artery disease, and elevated D-dimer level (>230 ng/mL) were risk factors for arterial thrombosis, and arterial thrombosis events were also associated with increased mortality (adjusted HR, 1.99; 95% CI, 1.65–2.40). Bleeding is less common than that of coagulation in patients with COVID-19. However, it can occur especially in high-risk groups, such as patients receiving anticoagulant therapy or those with immune thrombocytopenia (ITP) [37,38,39,40].

## 4. Diagnosis

Blood tests such as complete blood count, coagulation tests (PT and aPTT), fibrinogen, and D-dimer are mainly used to evaluate patients with COVID-19 and coagulation abnormalities because of the significant thrombocytopenia and lymphopenia [41]. Serial follow-up of D-dimer is not necessary because the purpose of D-dimer measurement is generally to evaluate the severity of the disease and treatment strategy changes based on clinical features rather than numerical changes in D-dimer. These tests may provide prognostic value and influence treatment decisions. An increase in D-dimer levels is associated with a worse prognosis [42]. Tang et al. reported that non-survivors had significantly higher D-dimer and FDP levels and longer PT and aPTT [1]. Fibrinogen and vWF antigen levels are significantly higher in patients with COVID-19 [41]. Escher et al. also observed a concurrent massive elevation of vWF accompanied by increased factor VIII clotting activity, which means that the increased vWF points are towards massive endothelial stimulation and damage [43]. In addition, endothelial activation markers, including soluble VCAM-1 (sVCAM-1), soluble TNF receptor I (sTNFRI), and heparan sulfate, were increased in patients with COVID-19, and the increased expression was related to COVID-19 disease severity [44]. Imaging tests are not routinely performed to screen for thromboses. This is because there is no evidence that these practices improve outcomes and may unnecessarily expose healthcare workers to an additional risk of infection. If there are symptoms or signs suggestive of DVT or PE, confirmation by imaging tests is necessary. Patients with suspected DVT should undergo ultrasonography. Normal levels of D-dimer are sufficient to rule out a diagnosis of PE, but elevated D-dimer levels are not specific to PE and are not sufficient to make a diagnosis. Computed tomography (CT) with pulmonary angiography is an essential test to confirm or rule out the diagnosis of patients with suspected PE based on unexplained hypotension, tachycardia, or worsening of respiratory status. Various other diagnostic tests, such as conventional angiography, compression ultrasonography with Doppler, and magnetic resonance imaging with venography, for various sites of thrombosis can also be considered. It is also important that other causes of thrombosis with or without thrombocytopenia be considered for differential diagnosis, including classic heparin-induced thrombocytopenia, DIC, immune thrombocytopenic purpura, thrombotic thrombocytopenic purpura, estrogen-containing drugs, hypersplenism, malignancy, trauma, surgery, pregnancy, immobility, and hereditary diseases such as thrombophilia [45].

## 5. Treatment

To date, several high-quality prospective randomized studies have supported standard treatment strategies for hypercoagulability [46,47,48,49,50,51]. Therefore, experts recommend that patients with COVID-19 who have prior thrombotic events or are highly suspected of TE should be managed with therapeutic anticoagulants regardless of whether they are critically ill or non-critically ill [46,47]. In addition, several studies have compared the effects of therapeutic-dose and prophylactic or intermediate-dose anticoagulants [46,47,48,49,50,51,52]. However, if we review studies comparing the effectiveness of therapeutic-dose and prophylactic-dose anticoagulants, most reported that therapeutic-dose anticoagulants do not significantly improve clinical outcomes compared with prophylactic-dose anticoagulants (Table 2). Lopes et al., Perepu et al., and Sholzberg et al. reported that therapeutic-dose anticoagulants did not improve clinical outcomes, such as applying invasive or non-invasive mechanical ventilation, admission to the ICU, and death compared with that of prophylactic anticoagulants, but rather increased the risk of bleeding [48,49,50]. On the other hand, Spyropoulos et al. reported that therapeutic doses of low-molecular-weight heparin (LMWH) could reduce thromboembolic events and death compared with that of prophylactic or intermediate-dose heparin treatment in high-risk patients with COVID-19 with high D-dimer levels [51]. The decision to use a full-dose (high-dose or therapeutic) anticoagulant should be based on a complex and comprehensive clinical evaluation. It is not recommended to change the anticoagulant based on isolated changes in some laboratory values, such as D-dimer. LMWH is generally recommended to be administered at a fixed dose (e.g., enoxaparin 1 mg/kg) without monitoring. In contrast, unfractionated heparin (UFH) is titrated based on laboratory tests such as aPTT and anti-factor Xa activity [53]. Standard indications for therapeutic dose include documented or strongly suspected VTE and clotting of vascular access devices unless there are restrictions on the use of anticoagulants or heparin. Severe COVID-19 is not considered an indicator of the therapeutic-dose anticoagulant. The choice and administration period of anticoagulants were similar to those of patients with DVT or PE due to acute medical illnesses other than COVID-19. As with patients without COVID-19 who have developed DVT or PE without persistent risk factors for VTE, therapeutic-dose anticoagulants can be discontinued after 3 months of treatment when COVID-19 has mostly recovered. Therapeutic anticoagulation is appropriate for patients with repeated coagulation of endovascular access devices (arterial and central venous catheters) despite anticoagulation with a prophylactic dose. It is also suitable for patients treated with continuous renal replacement therapy or ECMO. Some case series have reported that tPA improves mortality in patients with acute respiratory distress syndrome (ARDS) associated with COVID-19 [54,55,56]. However, tPA is not generally used for patients with nonspecific findings, such as simple hypoxia or hypercoagulability. It is suitable for specific indications, including limb-threatening DVT, massive PE, acute stroke, and acute myocardial infarction. Aspirin is not prescribed for patients with COVID-19 who have not been taking aspirin before, as it increases the risk of bleeding and does not improve COVID-19 outcomes. The role of antiplatelet agents other than aspirin is currently under investigation [57,58]. The established anticoagulation therapy for hospitalized patients with COVID-19 in several guidelines with high-quality evidence is summarized in Figure 2 [59,60,61,62]. The intensity of anticoagulation (prophylactic versus therapeutic dosing) should be individualized based on the patient’s thrombotic and bleeding risk. Several laboratory predictors of VTE in hospitalized patients with COVID-19 have been reported, including thrombocytopenia, elevated D-dimer levels, prolonged PT, and hypofibrinogenemia [63,64]. The choice of anticoagulant depends on the patient’s clinical condition and the anticipated need to discontinue anticoagulation. Among the available anticoagulants, direct oral anticoagulants (DOAC), such as apixaban, edoxaban, and ribavroxaban are the most preferred. In addition, oral direct thrombin inhibitors such as dabigatran, fondaparinux, danaparoid, argatroban, and bivalirudin can be considered. In the absence of active bleeding, with appropriate adjustments for body weight and renal function, the recommended maximal therapeutic dosing is appropriate. The appropriate duration of anticoagulation treatment is unknown. Therapeutic plasma exchange (TPE) is recommended for patients with CVT, multiple thromboses with severe thrombocytopenia (platelet count less than 30,000), or refractory disease [45,65]. Pavord et al. noted that TPE was associated with a 90% survival rate in patients with severe thrombocytopenia and CVT or extensive thrombosis, which led to strong consideration of TPE in these individuals [45]. Platelet transfusions are usually only considered for fatal bleeding, and platelet transfusions are minimized to prevent the exacerbation of thrombosis, with the exception of these indications. In other words, platelet transfusion should be offered to patients with life-threatening complications, including bleeding, or who require emergency surgery.

## 6. Prevention

CAC may be asymptomatic during the early phase of the disease and associated with a poor prognosis due to thrombosis occurring anywhere in the critical organs of the body [66]. In particular, D-dimer has already been identified as an independent poor prognostic factor and can provide valuable information on the severity of the disease [1,42,67]. Therefore, preventing thrombosis in patients with COVID-19 is considered an important method to improve the prognosis of the disease. Rentsch et al. showed that early initiation of prophylactic anticoagulation in admitted patients with COVID-19 could decrease the risk of 30-day mortality without increasing the risk of serious bleeding events [52]. Therefore, VTE prophylaxis is required for all hospitalized patients with COVID-19 unless the use of anticoagulants or heparin is contraindicated. In addition, all patients with COVID-19 in the ICU or non-ICU without suspected or documented VTE require prophylactic anticoagulation. Sometimes, patients with a high suspicion of VTE that cannot be documented may be treated with higher (therapeutic) doses of anticoagulants. In addition, patients already on high-dose (therapeutic) anticoagulants for other coagulation disorders should continue their treatment at the same therapeutic dose level unless there are contraindications such as active bleeding. The prophylactic doses of anticoagulants used were as follows. Enoxaparin is recommended to be 40 mg once daily for creatinine clearance (CrCl) of 30 mL/min or more and 30 mg once daily for CrCl of 15–30 mL/min. In Europe, dosing is based on units rather than milligrams. A typical prophylactic dose is 4000 units once daily, or 100 units/kg once daily [68,69]. Dalteparin is recommended at 50,00 units once daily, nadroparin at 3800 anti-factor Xa units once daily (BW under 70 kg) or 5700 units once daily (BW over 70 kg), and tinzaparin at 4500 anti-factor Xa units once daily. For patients with a CrCl < 15 mL/min or renal replacement therapy, unfractionated heparin, which is much less dependent on renal clearance, is recommended. LMWH is known to reduce the risk of VTE in hospitalized patients. Paranjpe et al. reported that anticoagulation was associated with improved in-hospital survival in intubated patients (71% in anticoagulated patients vs. 37% in non-anticoagulated) [70]. Bleeding events were not significantly different between the anticoagulated and non-anticoagulated patients (3% vs. 2%). Rentsch et at. reported that 84% of enrolled patients were started on prophylactic anticoagulant within the first 24 h of admission and found improved survival in the anticoagulated cohort (cumulative incidence of mortality at 30 days, 14.3% vs. 18.7% in those who did not receive prophylactic anticoagulant) [52]. Tang et al. reported that prophylactic enoxaparin (40–60 mg once daily) in patients with severe COVID-19 improved survival compared with that of controls in patients with high D-dimer levels (28-day mortality, 32.8% vs. 52.4%, *p* = 0.017) [71]. However, clinical trials have not demonstrated a consistent benefit with escalated doses. Studies have also been conducted to determine whether thromboprophylaxis is necessary for patients to recover from COVID-19 and be discharged. Ramacciotti et al. reported the potential benefits of platelet prophylaxis after discharge in high-risk patients in the MICHELLE trial [72]. In this trial, 320 hospitalized patients with COVID-19 at high risk of VTE but had no TE were randomly assigned. The group treated with 10 mg rivaroxaban daily for 35 days after discharge and the group without anticoagulants were compared. Symptomatic or fatal VTE, asymptomatic VTE on bilateral lower-limb venous ultrasound and CT pulmonary angiogram, symptomatic arterial TE, and cardiovascular death occurred in 3% of patients assigned to the rivaroxaban group and 9% of patients assigned to the without anticoagulant group (relative risk 0.33, 95% CI, 0.12–0.90; *p* = 0.0293). No major bleeding occurred in either group. The authors suggested that the use of extended prophylactic-dose rivaroxaban should be considered upon hospital discharge to improve the clinical outcomes in patients with COVID-19. Anticoagulants are generally not used in outpatients. Connors et al. reported that anticoagulants or aspirin are not needed in stable symptomatic outpatients with COVID-19 in the accelerating COVID-19 therapeutic interventions and vaccines (ACTIV)-4B trial, which was stopped early due to a very low number of events [73]. This trial did not reveal any clinical differences between the group receiving therapeutic and prophylactic doses of apixaban and the placebo group. Thus, outpatient thromboprophylaxis may be appropriate in selected individuals with COVID-19 who have thrombotic risk factors such as prior VTE or recent surgery, trauma, or immobilization. Therefore, outpatient thromboprophylaxis should only be considered for selected patients with COVID-19 who have prior VTE or recent surgery, trauma, or risk factors for thrombosis, such as immobilization. The bleeding did not appear to be a major manifestation of COVID-19. The approach to bleeding is similar to that in individuals without COVID-19 and may consider anticoagulant discontinuation, transfusions, or specific therapies, such as factor replacement.

## 7. Conclusions

CAC is associated with an inflammatory response that causes endothelial injury, resulting in a hypercoagulable state. These thrombotic events are frequent in hospitalized patients with COVID-19 and can significantly contribute to mortality and morbidity. Some mechanisms have been revealed to be potentially implicated in COVID-19 thrombosis. Recent randomized controlled trials have shown low mortality and great preventive effects by providing optimal treatment and prevention methods. Based on these results, early anticoagulant prevention is essential for all hospitalized patients with COVID-19.

## Figures and Tables

**Figure 1 microorganisms-10-01556-f001:**
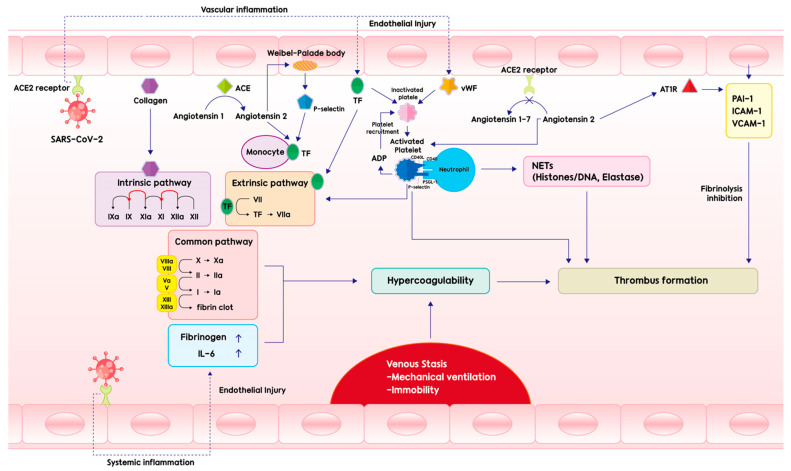
Pathophysiology of thrombosis in patients with COVID-19. Abbreviations: ACE, angiotensin-converting enzyme; TF, tissue factor; vWF, von Willebrand factor; ADP, adenosine diphosphate; ACE2 receptor, angiotensin-converting enzyme 2 receptor; AT1R, angiotensin II type 1 receptor; NETs, neutrophil extracellular traps; PAI-I, plasminogen activator-1; ICAM-1, intercellular adhesion molecule 1; VCAM-1, vascular cell adhesion molecule 1; IL-6, interleukin-6.

**Figure 2 microorganisms-10-01556-f002:**
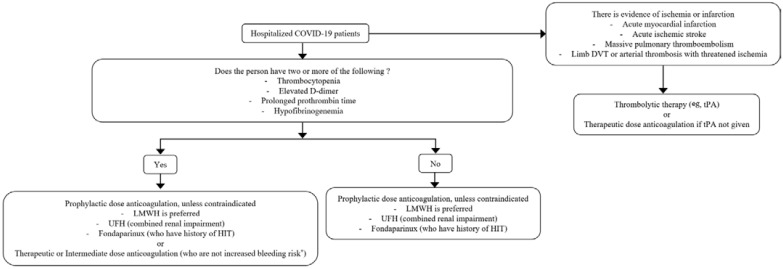
Anticoagulation therapy in hospitalized patients wCOVID-19 Abbreviations: LMWH, low-molecular-weight heparin; UFH, unfractionated heparin; HIT, heparin-induced thrombocytopenia; DVT, deep vein thrombosis; tPA, tissue plasminogen activator.* Contraindications for the use of therapeutic anticoagulants: platelet count < 50 × 10^9^/L, hemoglobin (Hgb) < 8 g/dL, the need for dual antiplatelet therapy, bleeding within the last 30 days that required an emergency department visit or hospitalization, and a history of bleeding.

**Table 1 microorganisms-10-01556-t001:** Differences in laboratory results between COVID-19-associated coagulopathy and disseminated intravascular coagulopathy.

.	COVID-19-Associated Coagulopathy	Acute Decompensated Disseminated Intravascular Coagulopathy	Vaccine-Induced Immune Thrombotic Thrombocytopenia (VITT)
Major finding	Thrombosis	Bleeding	Thrombosis
Platelet	Normal/decreased	Decreased	Decreased
PT/aPTT	Normal/prolonged	Prolonged	Normal/slightly increased
D-dimer	Increased	Increased	Increased
Fibrinogen	Increased	Decreased	Decreased
Factor VIII	Increased	Decreased	Increased
Fibrin degradation product (FDP)	Increased	Increased	Increased

**Table 2 microorganisms-10-01556-t002:** Comparison of the effectiveness of therapeutic-dose and prophylactic-dose anticoagulant therapy.

Study	Number of Participants		Mortality (%)		Venous Thromboembolism (%)		Organ Support Free Day		Major Bleeding (%)	
	Therapeutic	Prophylactic	Therapeutic	Prophylactic	Therapeutic	Prophylactic	Therapeutic	Prophylactic	Therapeutic	Prophylactic
[46]	534	564	37.3	35.5	6.4	10.4	1	4	3.8	2.3
[48]	311	304	11	8	7	10	N/A	N/A	3	1
[50]	228	237	1.8	7.6	0.9	2.5	25.8	24.1	0.9	1.7
[51]	129	124	19.4	25	10.9	29	N/A	N/A	4.7	1.6

## Data Availability

Not applicable.

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
