# Peer review of "Coronavirus Disease 2019-Associated Coagulopathy"

_microorganisms, 2022, doi:10.3390/microorganisms10081556_

Round 1
Reviewer 1 Report
This article did a comprehensive review of COVID-19 associated coagulopathy. Overall, the manuscript was well-written. I just have several minor suggestions.
1. Line 30-31, Please add the associated following this sentence “. The incidence of TE in patients with COVID-19 has been reported in various studies [3-5]”
2. Line 50-61, Please cite associated references.
3. In table 2, may add the comparison with VITT.
4. Please add a new paragraph to describe the prognosis of CAC.
Author Response
- Line 30-31, Please add the associated following this sentence “. The incidence of TE in patients with COVID-19 has been reported in various studies [3-5]”
à Answer : We have described information on the incidence of thromboembolism based on COVID-19 disease severity, highlighting that thromboembolism is more common in patients with severe than mild cases. (Line 31-35)
- Line 50-61, Please cite associated references.
à Answer : We've added a reference to that content. (Line 56)
- In table 2, may add the comparison with VITT.
à Answer : We have added characteristic laboratory results of vaccine induced immune thrombotic thrombocytopenia (VITT) and compared with CAC and DIC. (Table 2)
- Please add a new paragraph to describe the prognosis of CAC.
à Answer : There are no studies that independently evaluated the prognosis of CAC. Just, we've described the prevalence of CAC according to the severity of the disease, or poor prognostic factors that progressed to severe disease, such as d-dimer. (Line 31-35, 177-179, 277-282)

Reviewer 2 Report
Coronavirus disease 2019 (COVID-19)-associated coagulopathy is known to be associated at any anatomical site and to symptom patterns. No data are referred about the quantitative Real Time PCR test in these patients.
Author Response
Reviewer 2; Coronavirus disease 2019 (COVID-19)-associated coagulopathy is known to be associated at any anatomical site and to symptom patterns. No data are referred about the quantitative Real Time PCR test in these patients.
à Answer : We described with reference numerals 28 and 29 that CAC can occur extensively in all parts of the body and that CAC is associated with SARS-CoV-2 RNAemia, which in turn contributes to poor prognosis of COVID-19 (Line 129-138).
